# Pre- and Perioperative Inflammatory Biomarkers in Older Patients Resected for Localized Colorectal Cancer: Associations with Complications and Prognosis

**DOI:** 10.3390/cancers14010161

**Published:** 2021-12-29

**Authors:** Troels Gammeltoft Dolin, Ib Jarle Christensen, Astrid Zedlitz Johansen, Hans Jørgen Nielsen, Henrik Loft Jakobsen, Mads Falk Klein, Cecilia Margareta Lund, Stig Egil Bojesen, Dorte Lisbeth Nielsen, Benny Vittrup Jensen, Julia Sidenius Johansen

**Affiliations:** 1Department of Medicine, Copenhagen University Hospital, DK-2730 Herlev, Denmark; cecilia.margareta.lund.01@regionh.dk (C.M.L.); Julia.Sidenius.Johansen@regionh.dk (J.S.J.); 2Department of Gastrointestinal Surgery, Copenhagen University Hospital, DK-2650 Hvidovre, Denmark; Ib.Jarle.Christensen@regionh.dk (I.J.C.); hans.joergen.nielsen@regionh.dk (H.J.N.); 3Department of Oncology, Copenhagen University Hospital, DK-2730 Herlev, Denmark; Astrid.Zedlitz.Johansen@regionh.dk (A.Z.J.); Dorte.Nielsen.01@regionh.dk (D.L.N.); benny.vittrup.jensen@regionh.dk (B.V.J.); 4Department of Surgery, Copenhagen University Hospital, DK-2730 Herlev, Denmark; henrik.Loft.Jakobsen@regionh.dk (H.L.J.); mads.falk.klein@regionh.dk (M.F.K.); 5Department of Clinical Medicine, Faculty of Health and Medical Sciences, University of Copenhagen, DK-2200 Copenhagen, Denmark; Stig.Egil.Bojesen@regionh.dk; 6Department of Clinical Biochemistry, Copenhagen University Hospital, DK-2730 Herlev, Denmark

**Keywords:** ageing, CRP, colorectal cancer, complications, inflammation, IL-6, inflammatory biomarkers, surgery, YKL-40

## Abstract

**Simple Summary:**

Colorectal cancer is the second most common cancer worldwide, and the incidence increases with age. The primary treatment for localized disease is surgical resection. Biomarkers identifying older patients at risk of complications following surgery are desirable to create a more individualized treatment plan. The purpose of this study is to investigate if circulating proteins related to inflammation (CRP, Il-6, and YKL-40) can provide information about the risk of complications and survival in older patients undergoing resection, and, furthermore, to investigate if this relation is different in older patients as compared to younger patients. We investigated 401 patients with localized colorectal cancer and found that older patients (*n* = 210) had higher levels of preoperative inflammatory biomarkers compared to younger patients (*n* = 191). High levels were associated with major complications after resection in older, but not in younger, patients. This may be useful in the future to design more personalized treatment plans.

**Abstract:**

The association between pre- and perioperative inflammatory biomarkers, major complications, and survival rates after resection of colorectal cancer (CRC) in older patients is largely unknown. The aim was to investigate age-dependent differences in these associations. Serum CRP, IL-6, and YKL-40 were measured preoperatively and on the first and second day after resection of CRC (stages I–III) in 210 older (≥70 years) and 191 younger patients (<70 years). The results from the complications was presented as an odds ratio (OR, with a 95% confidence interval (CI)) with logistic regression. Results from the mortality rates were presented as a hazard ratio (HR, with a 95% CI) using Cox proportional hazards regression. The preoperative inflammatory biomarkers were higher in the older vs. the younger patients. The risk of complications was increased in older patients with a high preoperative CRP (OR = 1.25, 95% CI 1.03–1.53), IL-6 (OR = 1.57, 95% CI 1.18–2.08), and YKL-40 (OR = 1.66, 95% CI 1.20–2.28), but not in younger patients. Mortality was higher in younger patients with high preoperative YKL-40 (HR = 1.66, 95% CI 1.06–2.60). This was not found in older patients. Elevated preoperative inflammatory biomarkers among older patients were associated with an increased risk of complications, but not mortality. Preoperative inflammatory biomarkers may be useful in assessing the risk of a complicated surgical course in older patients with CRC.

## 1. Introduction

Colorectal cancer (CRC) is an aging-associated disease and is the second leading cause of cancer death, with 881,000 estimated deaths worldwide in 2018 [1]. As the population ages, the number of older patients with CRC is expected to increase [2,3]. Surgical resection of the tumor is the primary treatment for localized CRC. In general, surgery is considered safe, also for older patients, but the occurrence of post-operative complications can reduce their quality of life and overall survival (OS) [4,5]. Older patients are more prone to the negative effects of cancer and surgery than their younger counterparts [6], and biomarkers identifying those at risk of a negative outcome are needed for the planning of appropriate treatment.

Aging is often accompanied by a chronic low-grade inflammation (inflammaging). It is assumed to be one of the main forces driving aging and is a risk factor for morbidity and mortality in older patients [7,8]. However, little is known regarding preoperative inflammation and its association with the prognosis and complications in older patients undergoing resection for CRC.

Inflammation is one of the hallmarks of cancer [9]. A recent systematic review and meta-analysis has described a relationship between chronic inflammation, as measured by inflammatory circulating biomarkers (including C-reactive protein (CRP) and interleukin-6 (IL-6)) and an increased cancer incidence, including CRC [10]. Furthermore, the presence of systemic inflammation preoperatively, that is found in 20–40% of patients with CRC, is a marker of a poor prognosis [11]; high preoperative CRP and a modified Glasgow Prognostic Score (mGPS) are associated with postoperative infectious complications in patients resected for CRC [12,13].

The most used biomarker of inflammation is CRP. This protein is mainly produced by hepatocytes [14], whereas other circulating biomarkers of inflammation, such as IL-6 and YKL-40, are secreted by inflammatory, stromal, and cancer cells [15,16]. IL-6 is a multi-functional signaling protein that facilitates the inflammation cascade, as well as key pathways and processes in the cancer microenvironment and regulates many hallmarks of cancer. IL-6 stimulates the stromal desmoplasia, promotes tumor-induced immunosuppression, decreases apoptosis, promotes proliferation and angiogenesis, facilitates metastasis, and stimulates the production of CRP by the hepatocytes [15,17,18]. High plasma IL-6 is associated with a short OS in patients with metastatic CRC [17,19]. YKL-40 (also called chitinase-3-like-1 protein [CHI3L1]) is a glycoprotein that also plays a role in inflammation, as well as in the remodeling of the extracellular matrix, angiogenesis, metastasis, and protection against apoptosis [16,20]. In patients with various solid tumors, including CRC, high plasma YLK-40 is associated with a short OS [21,22].

No studies have described the prognostic value of pre- and perioperative inflammatory biomarkers in older, compared to younger, patients undergoing elective surgery for CRC. Here, we report results from a prospective biomarker study of patients who underwent curative intended resection for CRC, with the aim to test the hypothesis that elevated pre- and perioperative levels of three inflammatory biomarkers (CRP, IL-6, and YKL-40) have a stronger association with the risk of complications, CRC recurrence, and survival in older patients (≥70 years) than they do in younger patients (<70 years).

## 2. Materials and Methods

### 2.1. Patients

We used data from the Danish REBECCA study (“Biomarkers in patients with colorectal cancer—can they provide new information on the diagnosis, treatment efficacy, adverse events and prognosis?”), an open cohort study initiated in July 2014. We prospectively included patients with histologically verified or suspected CRC at stages I–III at the Department of Gastroenterology, Herlev and Gentofte Hospital, Denmark. The enrollment period spanned from July 2014 to March 2019. The patients were followed until February 2021 or until their death, whichever came first. Dates of death were linked with the Danish National Registry of Patients using the 10-digit civil registration number assigned to all Danish citizens at birth or immigration [23]. 

Exclusion criteria included neoadjuvant chemo- or radiotherapy, metastatic disease at the time of surgery, or a history of earlier cancer. Patients with tumor in polyps that were removed before resection that had no residual tumors in the resected material, patients with intraoperative signs of disseminated disease, and patients with preoperative infection or recent trauma, such as hip surgery within the previous month, were excluded as well.

### 2.2. Biomarker Analysis

Blood for biochemical analyses was drawn from all patients just prior to the planned surgical resection. In a subset of patients, postoperative blood samples were also collected at day 1 and 2. The blood samples were centrifuged at 2300 g at 4 °C for 10 min, and the serum was then aliquoted and stored at −80 °C. Biochemical analyses of all biomarkers were performed on the same serum sample from each patient. 

CRP was measured as high-sensitive CRP using a sensitive CRP Ultra ready-to-use liquid assay reagent by an immunoturbidimetric method on a fully automated chemistry analyzer (Kit-test SENTINEL CRP Ultra (UD), 11508 UD-2.0/02 2015/09/23). The measurement range was 0.3–640 mg/L. The intra- and inter-assay coefficients of variation (CVs) were 3% and <15%. Elevated CRP was defined as >10 mg/L.

IL-6 was measured using a high-sensitive enzyme-linked immunosorbent assay (ELISA) (Quantikine HS600B, R&D Systems, Abingdon, UK) in accordance with the manufacturer’s instructions. The lower limit of detection for IL-6 was 0.01 ng/L, and the intra- and inter-assay CVs were ≤8% and ≤11%, respectively. Elevated IL-6 was defined as >4.92 ng/L, the 95th percentile in healthy blood donors [24].

YKL-40 was measured using an ELISA (Quidel Corporation, San Diego, CA, USA) in accordance with the manufacturer’s instructions. The lower limit of detection for YKL-40 was 20 μg/L, and the intra- and inter-assay CVs were <5% and <6%, respectively. Elevated YKL-40 was defined as an amount higher than the age-corrected 95th percentile [25]. 

Albumin was determined using photometric testing (Siemens Atellica CH 930). The inter-assay CV was <5%. Hypoalbuminemia was defined as <35 g/L.

The modified Glasgow Prognostic score (mGPS) was calculated from levels of albumin and CRP. Patients with normal albumin and CRP, or with isolated hypoalbuminemia, had a score 0; patients with only elevated CRP had a score of 1; and patients with both elevated CRP and hypoalbuminemia were allocated to a score of 2 [26].

The measurement of inflammatory biomarkers was done blinded to patient characteristics and study outcomes and in accordance with the REporting recommendations for tumor MARKer prognostic studies (REMARK) [27,28].

### 2.3. Covariates

Clinical data were obtained from a chart review blinded to the analysis of the biomarkers. 

We used participant-reported information on smoking habits and alcohol consumption. High alcohol consumption was defined as an alcohol intake above 7 (women) and 14 (men) units per week (1 unit is equal to approximately 12 g of alcohol). The body mass index (BMI) was calculated at enrollment. We also calculated the Charlson comorbidity index (CCI) that was not adjusted by age. Other covariates included the ECOG performance status (PS), the American Society of Anesthesiologists physical status classification system score (ASA), the tumor location, and CRC stage according to the UICC TNM classification, 8th edition. The type of operation was obtained from a chart review, and cases with an intraoperative conversion from the laparoscopic approach to open procedure were registered as open surgery. 

Complications followingsurgery were obtained from a chart review and were cross-checked with the Danish Colorectal Cancer Group database, a validated database for the prospective registration of all patients in Denmark diagnosed with a primary cancer in the colon and/or rectum [29,30]. Discrepancies were evaluated by the primary investigator and a consultant surgeon. Complications following surgery were classified by the Clavien–Dindo classification [31], and major complications were defined as complications requiring surgical intervention under general anesthesia, intensive care unit management, or patient death (>Clavien–Dindo grade 3a). 

CRC recurrence was assessed by a chart review, including the review of scans performed and the pathology findings. The vital status and date of death were obtained from the Danish National Citizen Registry using the patients’ national Danish identification number.

### 2.4. Statistics

Results were reported in accordance with the REMARK guidelines [28]. Descriptive statistics were used for baseline clinical characteristics and CRP, IL-6, and YKL-40. Characteristics were analyzed for patients age <70 years and ≥70 years and were compared using chi-square tests for categorical data, or the Wilcoxon rank sum test for continuous variables. 

The ratios of inflammatory biomarker levels between the different categories of each clinical covariate were calculated by comparing the means of the biomarker on a log scale. The back-transformed mean differences were ratios. 

Complications were analyzed using logistic regression, scoring markers as a dichotomized variable or as a continuous variable (log was transformed to base 2 to approach a normal distribution). Goodness of fit tests were also performed. The results were presented as odds ratios (OR) with a 95% confidence interval (CI), and the discrimination was assessed by the area under the receiver operating characteristic (ROC) curve. 

The time to the event data, the OS, and disease recurrence were analyzed using the Cox proportional hazards model. CRP, IL-6, and YKL-40 were scored as dichotomized or continuous variables on either a dichotomized or continuous scale (log-transformed to base 2). The model assumptions of proportional hazards and linearity were assessed by martingale residuals. Relevant interactions between age and the associated markers were tested. The multivariate models were also assessed by a 10-fold cross validation. Results are presented as hazard ratios (HR) with a 95% CI.

The level of significance was set to 5%. All calculations were done using SAS (v9.4, SAS Institute, Cary, NC, USA) and R (R version 3.6.3 (29 February 2020).

### 2.5. Ethics Approval and Consent to Participate

All patients gave written informed consent. The study was performed according to the declaration of Helsinki. The REBECCA study protocol was approved by the Ethics Committee of the Capital Region of Denmark (VEK j.nr. H-2-2013-078) and the Danish Data Protection Agency (j. nr. HEH-2014-044, I-suite nr. 02771 and PACTIUS P-2019-614).

## 3. Results

### 3.1. Preoperative Charecteristics, Inflammatory Biomarkers, and the Age Groups

We included 749 consecutive patients with stage I–III CRC. After exclusion according to the predefined criteria, the cohort for the further analysis of inflammatory biomarkers consisted of 401 patients with an interquartile age range of 57 to 66 years at their inclusion in the age group of <70 years and a interquartile range of 72 to 80 years for the age group ≥70 years. The distribution of patients by age is shown in Appendix A. The patient flow is shown in Figure 1.

The median age of the patients was 70 years and 224 (56%) were men. Patients ≥70 years were significantly more comorbid, as shown by the elevated scores for both CCI (*p* < 0.0001), ASA (*p* = 0.0004), and PS (*p* < 0.0001). Adjuvant chemotherapy was given in 135 patients (34%) in accordance with national guidelines [32]. Patients ≥70 years had higher preoperative CRP (*p* = 0.0185), IL-6 (*p* < 0.0001), and YKL-40 (*p* < 0.0001) than younger patients (Table 1). Baseline characteristics are shown in Table 1. Histopathological characteristics and treatment characteristics are shown in Table 2. The distribution of patients by age, stage, and adjuvant chemotherapy is shown in Appendix A.

### 3.2. Preoperative Inflammatory Biomarkers in Relation to Clinical Covariates Stratified by Age Groups

Elevated inflammatory markers, particularly IL-6, were associated with most clinical covariates usually associated with an adverse prognosis, although not all were significant. There were differences between age groups, as only patients ≥70 years with UICC high-stage tumors, compared to low-stage tumors, had significantly higher CRP and IL-6 (stage I vs. stage II: CRP ratio 1.71, *p* = 0.009; IL-6 ratio 1.70, *p* = 0.0002). This was not found for the age group <70 years. Younger patients who were obese, overused alcohol, or were smokers had higher levels of inflammatory markers than those with a normal BMI, no/normal alcohol consumption, and who were non-smokers. This was not found in older patients (Table 3).

### 3.3. Preoperative Inflammatory Biomarker Levels and Major Complications to Surgery 

A total of 62 (16%) major complications were observed: 27 (14%) in the age group <70 years and 35 (17%) in the age group ≥70 years. There was no difference in the occurrence of major complications between the two age groups (*p* = 0.48) (Table 2). 

A univariate analysis of the entire cohort showed that the preoperative high levels of IL-6 and YKL-40, but not CRP, were associated with major complications (IL-6: OR = 1.33, 95% CI 1.12–1.59; and YKL-40: OR = 1.39, 95% CI 1.12–1.72) (Figure 2). We found a significant interaction between the biomarkers and the age group (*p* = 0.04). When the cohort was divided according to the age groups, the association between high levels of both IL-6 and YKL-40 and major complications remained only in the older age group, in both univariate and multivariate analyses. 

Univariate and multivariate analyses of the clinical covariates’ association with major complications are shown in Table 4. The multivariate analysis, including CCI, sex, age, and YKL-40 showed that higher YKL-40 was associated with major complications in patients ≥70 years (OR = 1.60, 95% CI 1.15–2.23), but not in patients <70 years (OR = 0.91, 95% CI 0.60–1.38). Similar results were found for CRP (≥70 years: OR = 1.25, 95% CI 1.01–1.54; <70 years: OR = 0.87, 95% CI 0.67–1.13) and IL-6 (≥70 years: OR = 1.61, 95% CI 1.19–2.17; <70 years: OR = 0.95, 95% CI 0.66–1.38). 

The univariate analysis of CRP, IL-6, and YKL-40 levels dichotomized into low/high levels, and their associations with major complications, showed that only high IL-6 and high YKL-40 in the older age group were associated with major complications (Appendix A). 

CCI, ASA, and PS performed similarly in the multivariate models, showing strong associations with complications (Appendix A). 

### 3.4. Perioperative Increases in Inflammatory Biomarkers and assocoation with Major Complicationsand Survival

Postoperative serum samples were collected in a subset of patients (postoperative day 1, *n* = 172; and postoperative day 2, *n* = 137). In this subset, 23 major complications and 17 deaths were registered. In both age groups, the inflammatory biomarkers increased from preoperative levels to the first and second postoperative days (*p* < 0.0001). This was independent of the age group for CRP, IL-6, and YKL-40 (*p* = 0.65, *p* = 0.24, and *p* = 0.61, respectively) and whether the resection was performed as a laparoscopic or open procedure (*p* = 0.20, *p* = 0.86, and *p* = 0.12, respectively). The increase in inflammatory biomarkers is shown in Figure 3. The increase in the biomarker levels, according to the tumor location, major complications, and the type of surgery (laparoscopic/open) is shown in Appendix A. A two-fold increase in IL-6 from preoperative levels to the first postoperative day was associated with an OR of 1.81 (95% CI 1.17–2.79) for the probability of major complications in patients <70 years, in contrast to an OR of 0.94 (95% CI 0.61–1.44) in patients >70 years (test for an interaction of IL-6 X age, *p* = 0.04). On postoperative day 2, a two-fold increase in IL-6 from preoperative levels to the second postoperative day was associated with major complications in both age groups (OR = 1.75, 1.24–2.46, *p* = 0.002). The change in YKL-40 levels from preoperative levels to the first postoperative day was not significant for the prediction of complications (OR = 1.38, 95% CI 0.91–2.10, *p* = 0.13). The change in YKL-40 from preoperative levels to the second postoperative day was associated with complications in both age groups (OR = 2.02, 95% CI 1.26–3.26, *p* = 0.004).

Increasing biomarker levels from preoperative levels to the first postoperative day were not associated with a shorter OS (CRP: HR = 0.85, 95% CI 0.64–1.14, *p* = 0.28; IL-6: HR = 0.91, 95% CI 0.67–1.23, *p* = 0.53); and YKL-40 (HR = 1.08, 95% CI 0.71–1.77, *p* = 0.71).

### 3.5. Correlations between Inflammatory Biomarkers

Spearman’s rank correlations between the preoperative levels of the inflammatory biomarkers in all 401 patients were highest for CRP and IL-6 (r = 0.64, *p* < 0.001). This was also found on postoperative day 1 (r = 0.57, *p* < 0.001, *n* = 172) and day 2 (r = 0.57, *p* < 0.001, *n* = 137). The highest correlation among postoperative samples was between IL-6 on day 1 and CRP on day 2 (r = 0.68, *p* < 0.001).

### 3.6. Preoperative Inflammatory Biomarkers and Overall Survival

The median follow-up time after resection for CRC was 60 months (25th–75th percentile: 41–71 months, reverse Kaplan–Meier method). During the follow-up period, 64 patients (16%) died: 18 (9.4%) in the age group of <70 years and 46 (21.9%) in the age group of ≥70 years.

Univariate Cox regression analyses (log-transformed continuous biomarkers) showed that preoperatively elevated CRP (HR = 1.19, 95% CI 1.04–1.35), IL-6 (HR = 1.31, 95% CI 1.11–1.55) and YKL-40 (HR = 1.37, 95% CI 1.11–1.68) were associated with decreased OS (Table 5). An mGPS score of 1, compared to 0, was also associated with reduced OS (HR = 1.93, 1.05–3.56). None of the biomarkers showed an interaction with the age group. Adding all three biomarkers into the same model showed that high IL-6 had the strongest association with decreased OS (HR 1.31, 95% CI 1.11–1.55). A univariate Cox analysis of clinical covariates showed that sex, histology, the tumor location, ASA, CCI, PS, major complications, and cardiovascular and lung disease were significantly associated with OS. A univariate analysis of CRP, IL-6, and YKL-40 levels dichotomized into low/high groups, and their association with OS, is shown in Appendix A. In multivariate analyses (including sex, age, CCI, PS, major complications, and each of the biomarkers tested separately) YKL-40 showed a significant interaction between biomarkers and age below or above 70 years (*p* = 0.03).

High YKL-40 was associated with poor OS in the younger patients (HR = 1.66, 95% CI 1.06–2.60) but not in the older patients (HR = 0.94, 95% CI 0.73–1.22) (Table 5). Multivariate analyses including the same clinical covariates, but with CRP or IL-6 instead of YKL-40, showed no association with OS for the biomarkers (CRP in the group <70 years: HR = 1.15, 95% CI 0.88–1.49; CRP in the group ≥70 years: HR = 1.06, 95% CI 0.90–1.26; IL-6 in group <70 years: HR = 1.06, 95% CI 0.78–1.44; and IL-6 in group ≥70 years: HR = 1.02, 95% CI 0.78–1.33). Kaplan–Meier plots for each biomarker in all patients, and in the two age groups separately, are shown in Figure 4.

### 3.7. Preoperative Inflammatory Biomarkers and Disease Recurrence

The recurrence of CRC was found in 51 (12.7%) patients during the follow-up period. None of the biomarkers were associated with recurrence in the univariate analysis of either the entire group or the two age groups (Table 6). Using biomarker values dichotomized into low/high groups showed no association with disease recurrence (Appendix A).

## 4. Discussion

We studied biomarker levels in 401 patients before and during the first two postoperative days after resection for localized CRC and found that older patients (≥70 years) had higher preoperative levels of the inflammatory biomarkers CRP, IL-6, and YKL-40 compared to younger patients. This is in line with previous studies that show that older subjects have high levels of CRP, IL-1 receptor antagonists, IL-6, IL-6 receptor antagonists, and IL-18 due to the aging process itself, as well as age-related multimorbidities [33,34]. Interestingly, we also found that high levels of CRP, IL-6, and YKL-40 preoperatively in the older, but not in the younger, age group were associated with an increased risk of major complications after surgery. High levels of YKL-40 were also associated with reduced OS for patients <70 years. To our knowledge, the impact of age on the association between inflammatory biomarkers and surgical outcomes has not been reported before. 

The incidence of major complications, defined as a Clavien–Dindo grade of >3a in the present study, was evenly distributed between the two age groups. Multivariate analyses (including the biomarkers, age, CCI, and sex) showed that the preoperative levels of the studied inflammatory biomarkers were significant associated with the risk of major complications only in the older patients. This suggests that both the inflammaging process and comorbidities contribute to the surgical morbidity in older patients undergoing surgery for CRC. Replacing CCI with one of the commonly used comorbidity scores, ASA or PS, showed similar results. However, it should be taken into consideration that all measures of comorbidity and PS have limitations in reflecting true patient capacity. The preoperative inflammation levels in the group <70 years may be explained by an inflammatory tumor–host response that did not lead to an increased surgical morbidity. Previous studies of patients undergoing CRC surgery [12,13] have reported a relationship between inflammatory biomarkers (preoperative mGPS and CRP) and infectious complications (Clavien–Dindo 2–5), whereas we found that high preoperative levels of IL-6 and YKL-40 (but not CRP or mGPS) were associated with major complications (Clavien–Dindo > 3a). This may be useful information since the severity of complications is associated with a negative impact on the quality of life and OS in patients with CRC [4,35,36,37]. 

We also found that increases in IL-6 and YKL-40 during the first and second days following surgery were associated with major complications. An association between postoperative complications and the early increase in IL-6 on the first postoperative day [38] and immediately after surgery [39] has been demonstrated, but the association has not been investigated for YKL-40. The relationship between postoperative systemic inflammation and major complications is expected, since many complications have an inflammatory component (e.g., anastomotic leak) and other studies have shown that high CRP on postoperative days three and four is associated with anastomotic leak [38,40,41]. Earlier detection of older frail patients at risk of major complications after resection for CRC is important, as delayed intervention might have fatal consequences for these patients. Our results indicate that IL-6 was superior to the routinely used CRP regarding the early detection of complications.

Although our study had a relative short follow-up period for some of the patients, we found an association between high preoperative levels of the inflammatory biomarkers and a reduced OS. This confirms the previously found association between systemic inflammation and poor survival rates in patients with different types and stages of cancer, including CRC [21,42,43]. In our multivariate models, including CRP, IL-6, and YKL-40 (the association between sex, age, PS, and major complications were tested with each inflammatory biomarker separately), the negative impact of systemic inflammation on OS was seen only in the group <70 years. Comorbidities and old age (both associated with low grade systemic inflammation) are major determinators for OS, and the cause of death in the older age group, after resection for CRC, is often not cancer related [6]. We hypothesize that our finding of increased levels of preoperative inflammatory biomarkers in the absence of comorbidities in the younger patients suggest that the tumor–host systemic response is the cause of the significant association with OS. No association was found between the biomarkers on postoperative days one and two and survival. However, others have reported that high CRP on days four and five, after surgery for CRC, is associated with a reduced OS [44].

No association was found between the investigated inflammatory biomarkers and disease recurrence, regardless of age. Inflammation is one of the hallmarks of cancer [9] and is involved in tumor progression and angiogenesis. IL-6 has been reported to modulate the extracellular matrix in the liver, the most common site of metastasis, resulting in the formation of a pro-metastatic niche [45]. Previous studies have shown associations between different systemic inflammatory markers, including lymphocyte–CRP ratios, IL-6, and disease-free survival [43,46]. However, these larger studies included patients with advanced diseases (stage IV diseases and patients treated with neoadjuvant chemotherapy and radiotherapy), whereas the present study included only patients with stage I-III diseases and neoadjuvant treatment was an exclusion criterion. There were, therefore, few events of recurrence and a lack statistical power to draw conclusions.

We found differences between the two age groups regarding the impact of clinical covariates on inflammatory biomarker levels. Lifestyle factors, such as smoking and an overuse of alcohol, were associated with high inflammatory biomarker levels in the age group of <70 years but not in the older age group. The proinflammatory effects of smoking have been described in younger populations [47,48], with smokers having higher IL-6 and CRP compared to non-smokers. The absence of high levels of inflammatory biomarkers in older smokers, compared to non-smokers, may be explained as a consequence of declining immune function (immunosenescence) with age [49] and generally higher levels of inflammation caused by comorbidities and aging, which could attenuate the difference.

The strengths of this study include its prospective design and a homogeneous cohort where patients treated with neoadjuvant chemoradiotherapy, or with a previous history of cancer, were excluded to avoid high levels of inflammatory biomarkers caused by neoadjuvant treatment or other cancers. The median age of our patients resembles that of patients with CRC in the general population and the cohort had the same degree of comorbidity, but it was slightly less burdened by polypharmacy compared to the nationwide register-based cohort study of patients with CRC in Denmark [50]. Furthermore, the data completeness, with very few missing values and no loss of follow-up data, was valuable. 

The limitations of our study include the single-center set-up that restricts the demographic and socioeconomic composition of the cohort. We have no information about socioeconomic factors. We acknowledged that these factors could have an impact on both short- and long-term outcomes, as reported by others [51,52]. Furthermore, the sample size and follow-up time did not allow us to investigate whether the cause of death was associated with the investigated biomarkers. An analysis of the perioperative changes in the inflammatory biomarkers could only be calculated in 44% of the patients, since not all patients had samples collected after the operation. This reduces the statistical power of the analysis of the changes in relation to OS. Furthermore, the postoperative systemic inflammatory response following surgery may be affected by the surgical procedure and the type of anesthesia used [53,54]. A recent meta-analysis found that anesthesia, consisting of the total intravenous anesthesia using propofol, was associated with a lower CRP after surgery compared to inhalational agents [55]. However, the studies in this meta-analysis were heterogeneous and included only two studies of patients undergoing surgery for CRC. These studies found no difference in IL-6 levels between total intravenous anesthesia and inhalational anesthesia 24 h after surgery [56,57]. The type of anesthesia was not assessed in the present study, and we have no data on the use of regional anesthesia or preoperative dexamethasone, which may also impact the systemic inflammatory response after surgery [58,59].

## 5. Conclusions

In this large prospective biomarker study of patients with CRC in stages I–III, we found that patients aged ≥70 years had higher preoperative levels of the circulating inflammatory biomarkers CRP, IL-6, and YKL-40 compared to patients <70 years. The pre- and perioperative circulating inflammatory biomarkers provided information on the risk of surgical complications after resection of CRC in older patients. This may be useful in the future to design more personalized treatment plans before surgery in older cancer patients with CRC and comorbidities.

## Figures and Tables

**Figure 1 cancers-14-00161-f001:**
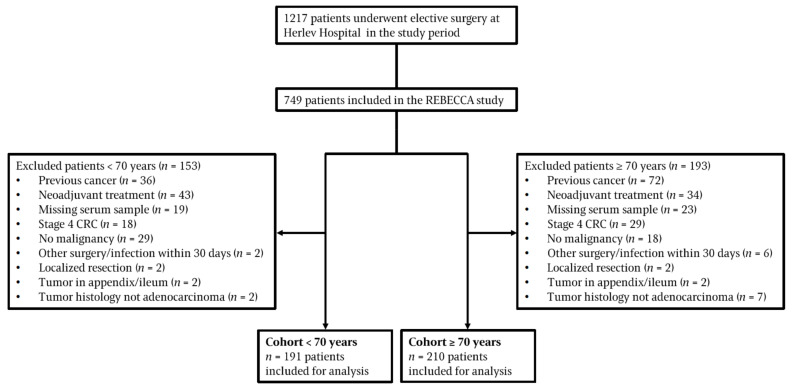
Patient flow. Abbreviations: CRC, colorectal cancer.

**Figure 2 cancers-14-00161-f002:**
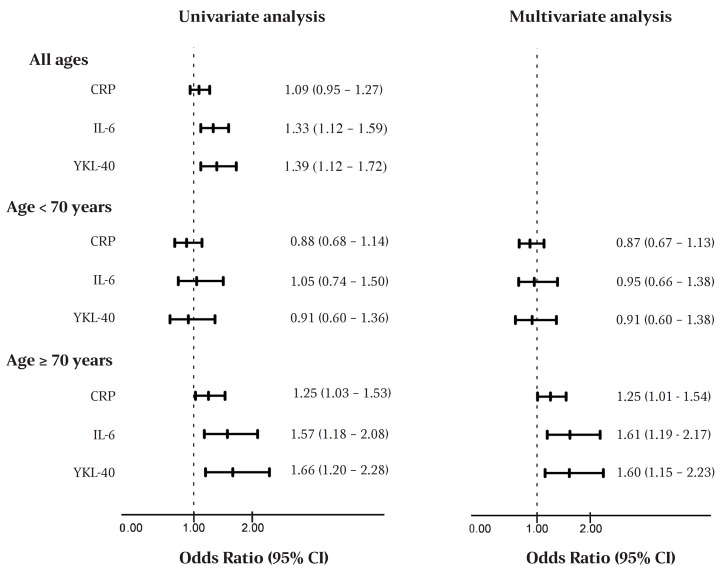
Forest plot illustrating inflammatory biomarker association with major complications, calculated as odds ratios. The multivariate analysis includes Charlson comorbidity index, sex, and age. Abbreviations: CRP, C-reactive protein; and IL-6, interleukin-6.

**Figure 3 cancers-14-00161-f003:**
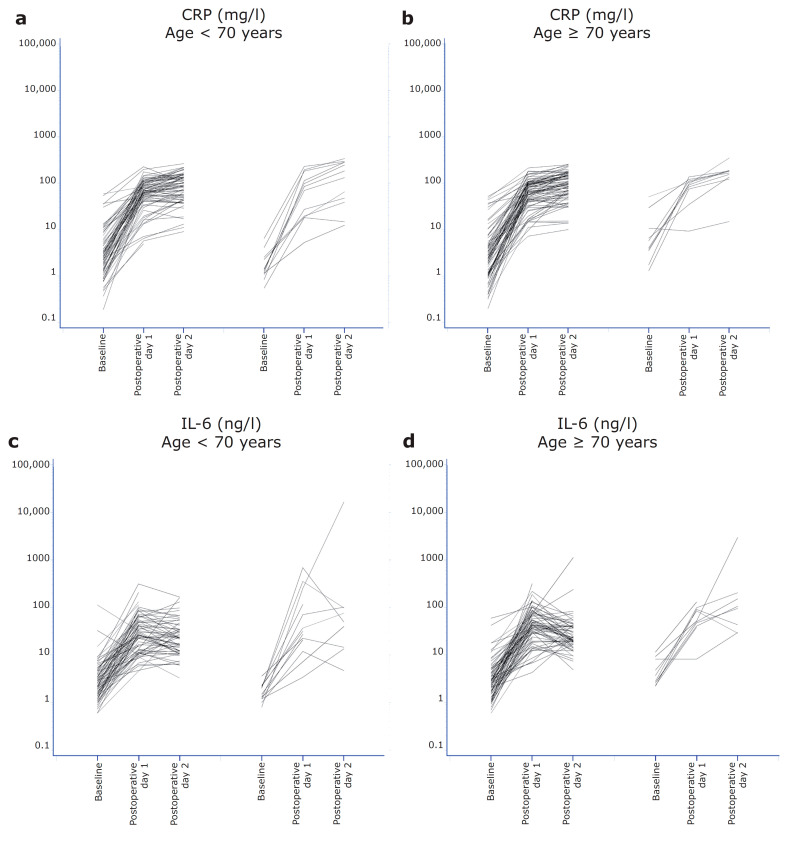
Spaghetti plots for biomarker levels (log scale) in pre- and postoperative serum samples. The figure illustrates patients with a surgical course without major complications (left) and patients with a surgical course with major complications (right). (**a**) CRP age <70 years. (**b**) CRP age ≥70 years. (**c**) IL-6 age <70 years. (**d**) IL-6 age ≥70 years. (**e**) YKL-40 age <70 years. (**f**) YKL-40 ≥70 years.

**Figure 4 cancers-14-00161-f004:**
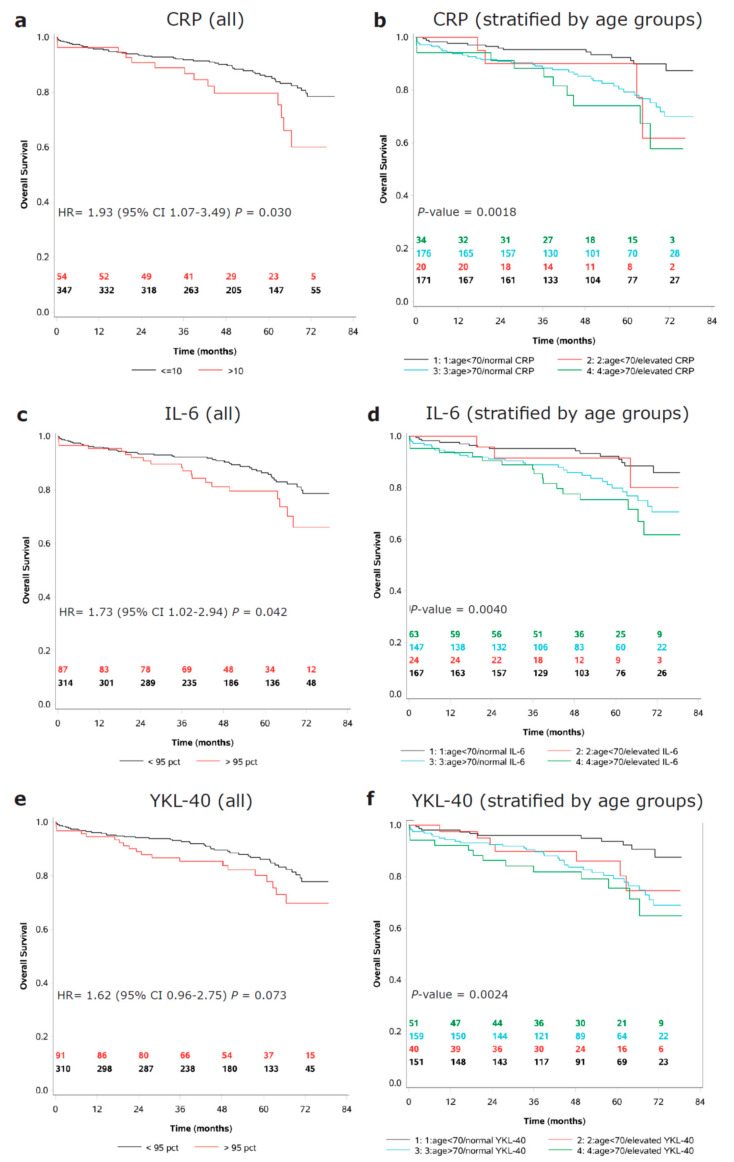
Kaplan–Meier plots showing preoperative dichotomized biomarker (normal/high) associations with overall survival. (**a**) CRP. (**b**) CRP stratified by age groups. (**c**) IL-6. (**d**) IL-6 stratified by age groups. (**e**) YKL-40. (**f**) YKL-40 stratified by age groups. Abbreviations: CRP, C-reactive protein; HR, hazard ratio; and IL-6, interleukin-6.

**Table 1 cancers-14-00161-t001:** Patient characteristics, *n* (%) *.

Characteristics	Variable	<70 Years (*n* = 191)		≥70 Years (*n* = 210)		*p*-Value
Age	median (range), years	63	(35–69)	75	(70–95)	
Sex	Male	105	(55)	119	(57)	0.7331
	Female	86	(45)	91	(43)	
Stage	I	60	(31)	61	(29)	**0.0312**
	II	58	(30)	89	(42)	
	III	73	(38)	60	(29)	
Performance status	0	167	(87)	125	(60)	**<0.0001**
	1	17	(9)	63	(30)	
	2	7	(4)	19	(9)	
	>2	0	(0)	3	(1)	
ASA score	I	89	(47)	59	(28)	**0.0004**
	II	88	(46)	123	(59)	
	III	14	(7)	28	(13)	
BMI ^a^	<18.5 kg/m^2^	5	(3)	8	(4)	**0.024**
	18, 5–25 kg/m^2^	69	(36)	101	(48)	
	>25–<30 kg/m^2^	66	(35)	67	(32)	
	≥30 kg/m^2^	49	(26)	32	(15)	
Smoking ^b^	Never	84	(44)	99	(47)	0.72
	Previous or current	102	(53)	104	(50)	
Alcohol consumption ^c^	Never or normal use	161	(84)	168	(80)	0.051
	Previous or current overuse	27	(14)	36	(17)	
CCI	0	143	(75)	105	(50)	**<0.0001**
	1–2	44	(23)	94	(45)	
	≥3	4	(2)	11	(5)	
Medications ^d^	0	65	(34)	31	(15)	**<0.0001**
	1–3	84	(44)	80	(38)	
	> 3	42	(22)	99	(47)	
Comorbidity	Cardiovascular disease	33	(17)	72	(34)	**0.0004**
	Lung disease	14	(7)	31	(15)	**0.018**
	Diabetes mellitus	14	(7)	26	(12)	0.092
Social status	Living alone	37	(19)	72	(34)	**0.0036**
mGPS	0	169	(89)	176	(84)	0.27
	1	18	(9)	31	(15)	
	2	2	(1)	2	(1)	
Biomarkers	CRP, median mg/L (range)	2.16	(0.2–64.7)	2.63	(0.2–70.4)	**0.0185**
	IL-6, median pg/L (range)	2.10	(0.6–112.8)	3.20	(0.6–59.6)	**<0.0001**
	YKL-40, median ug/L (range)	90	(20–790)	119	(27–1502)	**<0.0001**

* Unless stated otherwise. ^a^ BMI unknown: 4. ^b^ Smoking unknown: 11. ^c^ alcohol unknown: 9. ^d^ Medications unknown: 1. Abbreviations: ASA, American Society of Anesthesiologists physical status classification system; BMI, body mass index; CCI, Charlson comorbidity index; CRP, C-reactive protein; IL-6, interleukin-6; IQR, interquartile index; mGPS, modified Glasgow Prognostic score. Bold indicates statistic significant difference (*p*-value < 0.05).

**Table 2 cancers-14-00161-t002:** Histopathological characteristics and type of operation in relation to age.

Characteristics, *n* (%)	Variable	<70 Years		≥70 Years		*p*-Value
		*n* = 191		*n* = 210		
Tumor location	Right side	43	(23)	78	(37)	**0.0002**
	Left side	75	(39)	79	(38)	
	Rectum	72	(38)	45	(21)	
	Multiple tumors	1	(1)	8	(4)	
Tumor characteristics *	T = 1	28	(15)	18	(9)	
	T = 2	49	(26)	52	(25)	
	T = 3	96	(50)	110	(52)	
	T = 4	18	(9)	30	(14)	
	N = 0	118	(62)	150	(71)	
	N = 1	53	(28)	40	(19)	
	N = 2	20	(10)	20	(10)	
	V = 0	132	(69)	162	(77)	
	V = 1	49	(26)	45	(21)	
	V = 2	7	(4)	10	(5)	
	V = NK	3	(2)	0	(0)	
Histologic variants	Adenocarcinoma	176	(91)	188	(87)	0.44
	Mucinous	12	(6)	24	(12)	
	Signet ring cell	3	(2)	1	(1)	
	Medullary	3	(2)	4	(2)	
Differentiation	Low-grade differentiation	12	(6)	23	(13)	0.13
Surgery	Laparoscopic surgery	173	(90)	174	(83)	**0.0005**
Resection margin	R0	178	(93)	195	(93)	0.97
	R1 + 2	13	(7)	15	(7)	
Major complications	CD ≥ 3b	27	(14)	35	(17)	0.48
Adjuvant therapy	Receiving ACT	83	(44)	52	(25)	**<0.0001**
	No ACT, low stage	98	(51)	128	(61)	0.052
	No ACT, despite being indicated ** (percentage of patients with indication for ACT in each age group)	10	(11)	30	(37)	**<0.0001**
	Oxaliplatin-containing adjuvant treatment (percentage of all ACT given in each age group)	65	(78)	29	(56)	**0.006**

* Comparative analysis not performed due to low numbers. ** From histopathological findings. Values are presented as number (%) unless otherwise indicated. *p* < 0.05 was considered significant. Abbreviations: ACT, adjuvant chemotherapy; CD, Clavien–Dindo classification; N, lymph node; NK, not known; T, tumor; V, vein. Bold indicates statistic significant difference (*p*-value < 0.05).

**Table 3 cancers-14-00161-t003:** Preoperative inflammatory biomarkers in relation to clinical covariates in two age groups, <70 years and ≥70 years.

Clinical Covariates		<70 Years (*n* = 197)						≥70 Years (*n* = 210)					
		CRP		IL-6		YKL-40		CRP		IL-6		YKL-40	
Denominator	Numerator	Ratio *	*p*-Value	Ratio	*p*-Value	Ratio	*p*-Value	Ratio	*p*-Value	Ratio	*p*-Value	Ratio	*p*-Value
Male	Female	1.20	0.28	0.95	0.65	1.17	0.13	1.19	0.32	1.07	0.60	0.91	0.38
Stage: I	II	1.05	0.81	1.12	0.42	1.20	0.16	1.71	**0.0090**	1.70	**0.0002**	1.22	0.12
	III	1.18	0.42	1.04	0.78	1.03	0.81	1.72	**0.0016**	1.56	**0.0040**	1.01	0.96
Histology: Adenocarcinoma	Mucinous	1.72	0.12	0.94	0.81	1.14	0.54	1.28	0.39	1.11	0.60	1.26	0.21
	Signet cell	0.81	0.76	0.94	0.89	0.80	0.59	5.33	0.17	1.71	0.54	0.99	0.99
	Medullary	3.84	**0.047**	1.80	0.20	1.72	0.19	3.32	**0.056**	2.65	**0.027**	1.94	0.09
Location: Right	Left	0.86	0.51	0.76	0.060	0.90	0.42	0.65	**0.025**	0.72	**0.017**	0.77	**0.038**
	Rectum	0.73	0.16	0.72	**0.026**	0.66	**0.002**	0.51	**0.002**	0.72	**0.033**	0.80	0.094
BMI: normal	Low	2.36	0.11	1.82	0.093	1.04	0.90	0.82	0.67	1.19	0.59	0.60	0.070
	Overweight	1.27	0.23	1.23	0.12	0.99	0.93	0.87	0.50	0.93	0.62	1.01	0.93
	Obese	2.02	**0.0011**	1.60	**0.0011**	1.03	0.85	0.93	0.79	0.82	0.26	1.03	0.86
Tobacco: Never	Previous	1.18	0.37	1.31	**0.025**	1.01	0.93	0.91	0.63	0.92	0.52	1.25	0.053
	Current	1.64	**0.044**	1.99	**<0.0001**	1.46	**0.011**	0.83	0.50	0.88	0.51	1.01	0.93
Alcohol: Normal	Previous	1.74	0.29	1.41	0.32	1.24	0.49	1.68	0.36	0.86	0.86	0.83	0.60
	Overuse	2.06	**0.0061**	1.83	**0.0006**	1.78	**0.0003**	0.95	0.83	0.97	0.87	1.25	0.14
PS: 0	≥I	1.48	0.13	1.81	**0.0004**	1.57	**0.0034**	1.73	**0.0015**	1.71	**<0.0001**	1.46	**0.0005**
ASA: I	II	1.47	0.029	1.37	**0.0068**	1.16	0.16	1.43	0.070	1.36	**0.0021**	1.39	**0.0061**
	III	1.63	0.14	1.84	**0.0056**	1.52	**0.041**	1.98	**0.016**	2.11	**0.0001**	1.93	**0.0002**
CCI: 0	1–2	1.43	0.076	1.58	**0.0006**	1.10	0.43	1.34	0.095	1.21	0.12	1.21	0.083
	>2	1.06	0.93	1.99	0.074	2.39	**0.015**	1.18	0.68	1.49	0.14	1.43	0.14
Diabetes	Present	1.63	0.13	1.10	0.67	0.92	0.69	1.11	0.69	1.28	0.18	1.51	**0.010**
CVD	Present	1.03	0.89	1.41	**0.028**	1.06	0.67	1.09	0.66	1.23	0.11	1.17	0.19
LD	Present	1.25	0.49	1.26	0.29	1.03	0.87	1.08	0.75	1.07	0.67	1.11	0.49

* Ratios are calculated by comparing means of the biomarker on a log scale. Back-transformed mean differences are ratios. Legend: ASA, American Society of Anesthesiology; BMI, body mass index; CCI, Charlson comorbidity index; CVD, cardiovascular disease; LD, lung disease; and PS, performance status. Bold indicates statistic significant difference (*p*-value < 0.05).

**Table 4 cancers-14-00161-t004:** Clinical covariates and inflammatory biomarkers’ association with major complications.

Characteristics	Reference	Categories	Univariate (Adjusted for Age)				Multivariate (Adjusted for Age) *		
			OR	CI	*p*-Value	AUC	OR	CI	*p*-Value
Age	<70 years	≥70 y	1.10	0.84–1.45	0.48	0.52			
Sex	Male	Female	0.68	0.51–0.91	**0.010**		0.53	0.29–0.97	**0.03**
PS	0	≥1	1.49	1.13–1.98	**0.005**	0.59			
ASA	I	II	1.18	0.81–1.73	0.40				
		III	1.85	1.11–3.08	**0.019**	0.62			
Stage	I	II	1.13	0.78–1.64	0.51				
		III	0.98	0.66–1.44	0.91	0.53			
Location	left	Right	0.96	0.65–1.43	0.85				
		Rectum	1.43	0.99–2.07	0.059	0.58			
BMI	normal	Low	0.98	0.31–3.09	0.97	0.53			
		Overweight	1.12	0.66–1.93	0.67				
		Obese	1.03	0.56–1.88	0.88	0.93			
Smoking	never	Previous	0.94	0.49–1.82	0.86				
		Current	1.09	0.50–2.37	0.83	0.56			
Alcohol	normal	Previous	1.36	0.28–6.60	0.61				
		Overuse	0.97	0.43–2.17	0.99	0.51			
CCI	0	1–2	1.81	1.02–3.22	**0.043**		2.58	1.42–4.69	**0.0067**
		≥3	0.85	0.31–2.34	0.76	0.62	1.12	0.23–5.56	
Comorbidity	CVD	Present	1.59	1.19–2.12	**0.0017**				
	Lung disease	Present	1.60	1.12–2.28	**0.0101**				
	Diabetes	Present	1.41	0.96–2.07	0.0833				
mGPS	0	1	0.77	0.30–1.95	0.58				
		2	1.54	0.33–7.16	0.58	0.51			
CRP	CRP (all)		1.09	0.94–1.27	0.24	0.55			
	CRP < 70 years		0.88	0.68–1.14	0.33	0.60	0.87	0.67–1.13	0.301
	CRP ≥ 70 years		1.25	1.03–1.53	**0.027**	0.60	1.25	1.01–1.54	**0.038**
IL-6	IL-6 (all)		1.33	1.12–1.59	**0.0011**	0.60			
	IL-6 < 70 years		1.05	0.74–1.50	0.77	0.62	0.95	0.66–1.38	0.79
	IL-6 ≥ 70 years		1.57	1.18–2.08	**0.0021**	0.62	1.61	1.19–2.17	**0.002**
YKL-40	YKL-40 (all)		1.39	1.12–1.72	0.0028	0.60			
	YKL-40 < 70 years		0.91	0.60–1.36	0.63	0.64	0.91	0.60–1.38	0.66
	YKL-40 ≥ 70 years		1.66	1.20–2.28	**0.0020**	0.64	1.60	1.15–2.23	**0.005**

* In the multivariate analysis each inflammatory biomarker was included separately with regard to age, sex, and CCI due to the low number of complications. Values from the multivariate analysis of clinical covariates are shown using YKL-40 as biomarker. Abbreviations: ASA, American Society of Anesthesiologists physical status classification system; BMI, body mass index (low: <18.5; normal: 18.5–25; overweight: 25–30; obese: >30); CCI, Charlson comorbidity index; CRP, C-reactive protein; CVD, cardiovascular disease; IL-6, interleukin-6; mGPS, modified Glasgow Prognostic Score; Y, years; and PS, performance.Bold indicates statistic significant result (*p*-value < 0.05).

**Table 5 cancers-14-00161-t005:** Uni- and multivariate analyses of clinical characteristics, inflammatory biomarkers, and overall survival.

Characteristics	Reference	Categories	Univariate (Adjusted for Age)			Multivariate (Adjusted for Age)		
			HR	CI	*p*-Value	HR	CI	*p*-Value
Age	<70 years	≥70 y	2.47	1.43–4.26	**0.0012**	1.83	0.99–3.37	0.0531
Sex	Male	Female	0.42	0.24–0.73	**0.0023**	0.35	0.20–0.62	**0.0004**
PS	0	≥1	3.91	2.38–6.42	**<0.0001**	2.62	1.50–4.56	**0.0007**
ASA	I	II	1.66	0.90–3.06	0.10			
		III	5.12	2.53–10.4	**<0.0001**			
Stage	I	II	1.19	0.62–2.28	0.60			0.12 *
		III	1.64	0.87–3.09	0.13			
Location	Left	Right	1.97	1.11–3.52	**0.021**			
		Rectum	1.08	0.56–2.06	0.82			
Histology	Adenocarcinoma	Mucinous	2.74	1.43–5.26	**0.0024**			
		Signet Cell	2.32	0.32–16.84	0.41			
BMI	Normal	Low	1.58	0.48–5.21	0.45			
		Overweight	0.96	0.55–1.69	0.89			
		Obese	0.95	0.49–1.88	0.89			
Smoking	Never	Previous	0.83	0.48–1.46	0.52			
		Current	1.40	0.73–2.69	0.31			
Alcohol	Never/normal	Previous	1.50	0.36–6.15	0.57			
		Overuse	0.96	0.49–1.89	0.90			
CCI	0	1–2	3.03	1.80–5.12	**<0.0001**			
		≥3	4.74	1.80–12.49	**0.0016**			
Comorbidity	CVD	Present	3.28	2.00–5.38	**<0.0001**			
	Lung disease	Present	2.24	1.24–4.06	**0.0076**			
	Diabetes	Present	1.98	1.03–3.79	**0.039**			
Major	Complication	Present	3.35	2.00–5.61	**<0.0001**	2.43	1.42–4.16	**0.0012**
ACT	No	Yes	0.70	0.40–1.20	0.19			
mGPS	0	1	1.93	1.05–3.56	**0.035**			
		2	1.99	0.27–14.4	0.50			
CRP	CRP (all ages)		1.19	1.04–1.35	**0.009**			
	CRP < 70 years		1.10	0.85–1.43	0.48			
	CRP ≥ 70 years		1.18	1.02–1.37	**0.028**			
IL-6	IL-6 (all ages)		1.31	1.11–1.55	**0.0016**			
	IL-6 < 70 years		1.32	0.96–1.81	0.085			
	IL-6 ≥ 70 years		1.21	0.98–1.50	0.075			
YKL-40	YKL-40 (all ages)		1.37	1.11–1.68	**0.0035**			
	YKL-40 < 70 years		1.86	1.19–2.90	**0.0063**	1.66	1.06–2.60	**0.0275**
	YKL-40 ≥ 70 years		1.13	0.88–1.46	0.33	0.94	0.73–1.22	0.65

* *p*-value to include covariate in model. Abbreviations: ACT, adjuvant chemotherapy; ASA, American Society of Anesthesiologists physical status classification system; BMI, body mass index (low: <18.5; normal: 18.5–25; overweight: 25–30; obese: >30); CCI, Charlson comorbidity index, CRP, C-Reactive Protein; IL-6, interleukin-6; mGPS, modified Glasgow Prognostic Score; and PS, performance status. Bold indicates statistic significant result (*p*-value < 0.05).

**Table 6 cancers-14-00161-t006:** Univariate analysis of clinical characteristics and biomarkers’ association with disease recurrence.

Characteristics	Reference	Categories	Univariate		
			HR	95% CI	*p*-Value
Age	<70 years	≥70 years	1.46	0.83–2.57	0.18
Sex	Male	Female	0.70	0.40–1.24	0.22
PS	0	≥1	1.60	0.90–2.84	0.11
ASA	I	II	1.75	0.91–3,35	0.09
		III	2.83	1.17–6.84	**0.02**
Stage	I	II	6.77	1.56–29.5	**0.01**
		III	17.35	4.16–72.3	**<0.0001**
Location	Left	Right	1.62	0.85–3.09	0.14
		Rectum	1.00	0.49–2.03	0.99
Resection margin	R0	R1	3.91	1.83–8.37	**0.0004**
		R2	8.26	2.9–23.2	**<0.0001**
Histology	Adenocarcinoma	Mucinous	1.39	0.55–3.51	0.48
		Signet Cell	4.80	1.16–19.9	**0.03**
BMI	Normal	Low	2.64	0.77–9.00	0.12
		Overweight	1.63	0.86–3.08	0.14
		Obese	1.31	0.60–2.86	0.50
Smoking	Never	Previous	1.28	1.70–2.33	0.42
		Current	1.25	0.55–2.83	0.59
Alcohol	Never/normal	Previous	1.64	0.40–6.76	0.50
		Overuse	0.65	0.26–1.65	0.37
CCI	0	1–2	1.99	1.13–3.51	**0.02**
		≥3	2.51	0.76–8.34	0.13
Comorbidity	Cardiovascular disease	Present	2.21	1.25–3.94	**0.007**
	Lung disease	Present	1.67	0.79–3.56	0.18
	Diabetes	Present	1.82	0.85–3.87	0.12
Major	Complications	Present	1.25	0.59–2.66	0.56
ACT	Yes	No	2.23	1.29–3.87	**0.004**
mGPS	0	1	1.36	0.64–2.90	0.42
		2	2.13	0.29–15.5	0.46
CRP	CRP (all ages)		1.05	0.90–1.22	0.53
	CRP < 70 years		1.22	0.97–1.55	0.10
	CRP ≥ 70 years		0.94	0.77–1.14	0.52
IL-6	IL-6 (all ages)		1.14	0.93–1.40	0.19
	IL-6 < 70 years		1.11	0.78–1.56	0.57
	IL-6 ≥ 70 years		1.12	0.86–1.46	0.40
YKL-40	YKL-40 (all ages)		1.05	0.82–1.35	0.68
	YKL-40 < 70 years		1.21	0.80–1.84	0.37
	YKL-40 ≥ 70 years		0.92	0.66–1.27	0.60

Abbreviations: ACT, adjuvant chemotherapy; ASA, American Society of Anesthesiologists physical status classification system; BMI, body mass index (low: <18.5; normal: 18.5–25; overweight: 25–30; obese: >30); CCI, Charlson comorbidity index; CRP, C-reactive protein; IL-6, interleukin-6; mGPS, modified Glasgow Prognostic Score; PS, performance status; and R, resection margin status. Bold indicates statistic significant result (*p*-value <0.05).

## Data Availability

For data supporting the results of this study, contact the corresponding author.

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
