# Peer review of "Pre- and Perioperative Inflammatory Biomarkers in Older Patients Resected for Localized Colorectal Cancer: Associations with Complications and Prognosis"

_cancers, 2021, doi:10.3390/cancers14010161_

Round 1

Reviewer 1 Report

Dear author, I must congratulate you on the idea of ​​trying to clarify the management of elderly patients. I believe that it is an increasingly frequent topic in the management of any pathology and it is important to know what our attitude should be towards them.

I must say that I am surprised that throughout the work you say compares two groups of patients: young and old.

Analyzing the groups, it is seen that the group of elderly patients does not exceed 80 years. Could you justify it? This is not defined as an exclusion criterion; that patients are older than 80 cannot be included in it, right?

Next, I believe that the groups cannot be compared but rather a description of the two groups can be made. (Cohort groups). These are groups of patients with very different characteristics in terms of: morbidity, initial diagnostic stage and also in terms of subsequent management: the type of adjuvant treatment is different according to age and also the indication for its performance. The latter are already the characteristics of elderly patients and make them different from young ones; could not be a confounder?

A very high number of data is handled. Could you make a summary of the most significant? And how can it be applied in daily life? What do the authors advise? Should we make any changes to our activity based on the data found in your study?

As it says at the end of the discussion, it is understood that everything is based on 44% of the analyzes obtained? This is not too little to be able to reach conclusions?

Thank you very much for the effort and I would appreciate your clarification.

Sincerely

Reviewer 2 Report

CRC Review:

This manuscript is well written; the study is through and needs only minor edits. This study was to investigate if CRP, IL-6 and YKL-40 which are the circulating proteins related to inflammation could provide information about risk of complications and survival in older patients undergoing resection for colorectal cancer(CRC) versus younger patient population in the same disease. The sample size for this study was 401 (210 older pts & 191 younger pts). The primary objective was to find age-dependent differences in these associations. High inflammatory biomarkers preoperative are associated with postop infectious complications. This study is unique as none other studies have evaluated the prognostic value of pre and perioperative inflammatory biomarkers in older patients compared to younger CRC patients undergoing elective surgery. Figure 4 KP curves stratified by age groups and biomarkers is very informative.

 Recommendations/questions:

  1. Please state the reason for excluding neoadjuvant chemo. or radiotherapy metastatic disease
  2. Add citation for REMARK guidelines in the statistics section
  3. In the 12.7% recurrent CRC patients identified in the follow-up period what may be the reason that none of the biomarkers were associated with recurrence in the analysis?

Round 2

Reviewer 1 Report

Dear author, congratulations on the work done.
It is becoming more and more important to know data that help us to manage elderly patients.
Within your study I would like to know more details:

- The inclusion / exclusion criteria do not take into account the morbidity of the patients. Shouldn't it be a reason that the patients did not have any intercurrent disease that affected the elevation of inflammatory parameters?

- The age considered as elderly patients is very young: more and more patients over eighty years are those who arrive in good condition to be operated on. Are there differences between patients of 70 years and those of more than 80?

  • I find the analysis of the type of neoplasm that patients have to be lacking. Is it the same neoplasm of the right colon as the left? Also the type of surgery carried out: laparoscopy or not, anastomosis or not, .... I think these are data that can also affect complications and therefore may be study biases.

Congratulations

Sincerely